

# A small volume multiplexed pumping system for automated, high frequency water chemistry measurements in volume-limited applications

Bryan M. Maxwell[1], François Birgand[1], Brad Smith[2], Kyle Aveni-Deforge[1]

[1]Biological and Agricultural Engineering, North Carolina State University, 3110 Faucette Drive, Raleigh, 27695, USA
[2]Sungate Design Group, 905 Jones Franklin Road, Raleigh, 27606, USA

*Correspondence to*: Bryan M. Maxwell (bmmaxwel@ncsu.edu)

**Abstract.** An automated multiplexed pumping system (MPS) for high frequency water chemistry measurements at multiple

locations was previously reported. This technology showed potential to increase spatial and temporal resolution of data and improve our understanding of biogeochemical processes in aquatic environments and at the land-water interface. The design of the previous system precludes its use in volume-limited applications where highly frequent measurements requiring large sample volume would significantly affect observed processes. A small volume MPS was designed to minimize sample volume while still providing high frequency data. The system was tested for cross contamination between multiple sources and two

applications of the technology are reported. Cross contamination from multiple sources was shown to be negligible when using recommended procedures. Short-circuiting of flow in a bioreactor was directly observed using high frequency porewater sampling in a well network, and the small volume MPS showed high seasonal and spatial variability of nitrate removal in stream sediments, enhancing data collected from *in situ* mesocosms. The results show it is possible to obtain high frequency data in volume-limited applications. The technology is most promising for observing pore water solute dynamics and

improving existing solute transport models for saturated or partially saturated soils and media.

## 1 Advancements in high frequency water quality monitoring

Recent UV-vis field spectrometers provide an opportunity for high frequency *in situ* monitoring to increase temporal resolution of water quality data. Water quality has been measured with these instruments by correlating absorbance with chemical concentration (Crumpton et al. 1992; Suzuki and Kuroda 1987; Finch et al. 1998; Johnson & Coletti 2002; Rochelle-Newall

& Fisher 2002; Saraceno et al. 2009). Rieger et al. (2006) and Torres & Bertrand-Krajewski (2008) have measured total suspended solids (TSS) in wastewaters using partial least squares regression (PLSR) to correlate absorbance fingerprints to concentrations. Etheridge et al. (2014) expanded the technique to measure light- and non-absorbing constituents including $PO_4^{3-}$, total phosphorus (TP), nitrate ($NO_3^-$), total Kjeldahl nitrogen (TKN), TSS, dissolved organic carbon (DOC), and salinity in a brackish North Carolina marsh. Birgand et al. (2016) have shown that the technique might also be used to measure iron

and silica in lakes and reservoirs.



Among the spatial heterogeneity of biological and chemical processes in the environment (Merill & Tonjes, 2014; Kahlert et al., 2002; Dent & Grimm, 1999; Parkin, 1987), patches referred to as 'hot spots' are particularly interesting (e.g. riparian buffers, hyporheic zones). These 'hot spots' have been observed to have a disproportionate impact on biogeochemical cycles, and can be particularly active over short periods of time referred to as 'hot moments' (McClain et al., 2003; Vidon et al., 2010).

While temporal resolution has and will provide invaluable information at a particular monitoring station (e.g. Etheridge et al., 2014), expanding resolution to spatial data could illuminate tightly coupled processes and would greatly magnify the value of these instruments. Documenting the short-term fate of reactive nutrients within identified 'hot spots' might provide new insight into nutrient cycles and their controlling factors.

## 1.1 Increasing resolution of temporal *and* spatial data

An automated large volume multiplexed pumping system (MPS) capable of pump water from up to 12 sources to a UV-vis spectrophotometer was previously reported (Birgand et al., 2016). The 3.18 mm inside diameter (ID) tubing yields an average volume of 100 mL of water per sample, which is very well adapted when the source can easily accommodate such withdrawal without consequences on the source or the processes studied. Certain applications, however, may require minimizing sample withdrawal to avoid disturbing the observed process while still obtaining high frequency data. Observing solute dynamics in

soils, particularly those with low drainable porosity, is one such example of a volume-limited application.

In this article, we are reporting the capabilities offered by a small volume MPS coupled with a field spectrophotometer to obtain high resolution water quality data in both time and space. This system can accommodate small volumes (< 15 mL) from up to 12 different sources located within a 10 m distance of the instrument and provide absorbance measurements at better than hourly intervals at each location. This system was designed to minimize inline volumes for volume-limited

applications. This article describes the instrument and the challenges involved, evaluates its performance, and reports two applications of this system in such volume-limited conditions.

## 2 Small volume multiplexed pumping system

The principle followed for this instrument is to use a portable spectrophotometer as a central portable laboratory, coupled with the MPS able to sequentially pump water from several points within a reasonable distance of the probe. The small volume

MPS is designed for high frequency sampling in volume limited applications. The solution to minimizing sample volume is small diameter tubing and a low volume flow-through cell. The challenges of this design include clogging of tubing and high head losses due to surface tension forces in the tubing. The latter implies higher pump suction and tends to generate sample residuals left after purging along the length of the tubing, increasing the potential for cross contamination between samples. This drawback demands specific evaluation of the small volume MPS before use.

### 2.1 System Design and Hardware





The major components of the system include: 1) a UV-vis spectrophotometer fitted with a flow-through cuvette, 2) a bidirectional peristaltic pump, 3) small diameter Polytetrafluoroethylene (PTFE or Teflon) tubing, 4) an Arduino control board (Arduino, www.arduino.cc), 5) a 12 port valve, 6) a 3 way valve manifold, and 7) an optional fractional volume collector.

The general automated 'sampling sequence' for the system starts with pumping from source $n$ to a flow-through
spectrophotometer cuvette via 0.9 mm ID PFTE tubing and a peristaltic pump. After the water has entered the cuvette the spectrophotometer takes a measurement and water is then purged back either to the source, to a fractional volume collector, or to waste using a 3 way valve manifold. The multi-port valve then selects source $n+1$ and a similar 'sampling sequence' takes place. Up to 12 'sampling sequences' corresponding to the 12 ports of the multi-port valve can occur during a 'sampling cycle'. A fractional volume collector can be used to collect selected sample volumes so that readings by the spectrophotometer
can be compared to lab measured concentrations.

Control of the MPS pumping, purging, valve actuation, and activity logging is performed by an Arduino Mega 53 pin control chip (Arduino, www.arduino.cc). This inexpensive control board utilizes open-source software for easy-to-use programming and hardware interfacing, although other similar board/microprocessors can be used. The high frequency water quality probe used is a UV-vis field spectrophotometer Spectro::Lyser™ by s::can™ fitted with a 4 mm path length, 1.1 mL flow-through
quartz cuvette (Starna Cells, Inc. model 46-Q-4) placed vertically to facilitate cuvette drainage by gravity during purging. The water quality probe is used as a 'master' instrument to dictate the MPS 'slave' system when to begin a 'sampling sequence'. We use an existing capability of the spectrophotometer to send a 12 V signal at an adjustable time prior to a measurement as a trigger for the MPS, although any other triggering method (e.g. time-based using the Arduino control board) could be used.

A peristaltic pump provides pumping and purging at 20 mL min$^{-1}$ flow rate acceptable for the 0.9 mm ID tubing used. A
MOSFET h bridge controlled by the Arduino reverses DC polarity at the pump to alternate between pumping and purging. The 12 port valve used is a low pressure Cheminert® 12 position valve from Valco Instruments. The multi-port valve advances to the next desired port using a pneumatic actuator powered by compressed air (bottle or 12 V compressor). Although the valve advances though ports sequentially, the user can select the ports to use with the Arduino controller. Three Arduino-controlled 3 way solenoid valves (Takasago MTV series) are used for different sampling and purging configurations (Sec 2.2).
The Arduino board is fitted with an SD card reader/writer for activity logging, an RTC clock for time-keeping, an LCD panel for system output display, and operable switches for manual control.

The system has been designed for unattended operation for long periods of time (e.g. days or weeks), although it is limited by the spectrometer data storage, fouling of the cuvette, and battery power (and sample storage of the fractional volume collector if used). The time for a 'sampling sequence' is limited by the longest pumping time from furthest source. Ultimately this
determines the frequency of spectrometer measurements since the probe has a set time interval between measurements. The sampling frequency for each source depends on the 'sampling sequence' time intervals and on the number of sources (1 to 12).

## 2.2 System Configurations

Several configurations have been programmed to increase versatility of the instrument and minimize cross contamination between samples (Figure 1) by actuating one or more of the 3 way valves using the Arduino control board.

### 2.2.1 Pumping from source to probe and purging back to source.

When it is necessary to purge all sample volume back to the source, the system is configured as Fig. 1(a) during pumping (downward arrow on the peristaltic pump) and as Fig. 1(b) during purging (upward arrow on the peristaltic pump). To measure absorbance in the 1.1 mL cuvette used, a minimum sample volume of ~7 mL is required which includes cuvette volume ($V_{cuvette}$) plus inline tubing volume. The system is purged by air by running the pump in reverse. To minimize cross contamination without using a DI rinse, the cuvette can be rinsed with additional sample volume >7 mL and temporarily stored in the post-cuvette storage volume.

### 2.2.2 Purge to waste

In this configuration, it is not desired that the sample be purged to the source, so it is purged to waste (Fig. 1(c)).

### 2.2.3 Pump to waste

This configuration is used for purging residuals of the sample drawn from the previous source *n -1* that remain in the tubing and rinsing with sample volume from the current source *n* (Fig. 1(d)). After the estimated time for new water from the current source to reach the waste valve has elapsed, the system switches to configuration in Fig. 1(a) for sample measurement by probe.

### 2.2.4 Fractional volume collection (FVC)

This configuration is used for collecting measured sample volumes to compare values given by the probe to later lab analysis (Fig. 1(e)). The actual sequence consists in purging the FVC tubing with air using Fig. 1(e) configuration but in the pumping mode (downward arrow on the peristaltic pump, not shown) until water has passed the 3 way valve closest to the FVC, then use Fig. 1(c) to purge any cross contaminated water to waste before samples are sent to the FVC using configuration in Fig. 1(e). Any water left in tubing after the FVC sample vial is full is purged to waste.

### 2.2.5 DI Water Rinse

Between each sample, the user may choose to rinse the tubing and cuvette with DI water to create consistent cross contamination between samples and residual DI water droplets (Fig. 1(f)). This configuration draws water in from a DI water source and pumps it though the lines and into the cuvette. Consistent cleaning of the quartz cuvette decreases optical fouling over time. The cuvette and post-cuvette storage volume are rinsed and the water is then purged to waste.

### 2.3 System Testing

In laboratory continuous flow systems, the risk for cross contamination between consecutive samples is mechanically minimized by having unidirectional flow and by rinsing the lines with a carrier liquid between samples. Our system has been




designed to allow for bidirectional flow, purging the lines instead with air. Because of surface tension forces purging with air can leave micro-droplet residuals in the lines, opening the possibility for cross contamination between consecutive samples. This is particularly problematic since sample volumes needed for analysis each time are by design small. Although this is admittedly undesirable, this can still be acceptable as long as the risks be evaluated and solutions to minimize the risks be

known.

Cross contamination could arise from two processes. First, cuvette contamination could occur when a water sample inside the cuvette during instrument reading is contaminated by droplets from the previous sample, either in the cuvette or in the MPS tubing. Second, cross contamination of the source, or source contamination, could occur when the 'purging back to source' configuration is used (Fig. 1(b)) and the source itself is contaminated by residuals from other sources. Both contamination

possibilities were evaluated in two separate experiments. A third test of performance quantified the relationship between sampling frequency and the distance from the source to the system, since increased pumping time for a 'sampling sequence' leads to lower data frequency.

Methods for evaluating cuvette contamination and source contamination were identical to those used in Birgand et al. (2016). Alternating measurements between sources of high and low $NO_3^-$ concentration ($[NO_3]_{high}$ and $[NO_3]_{low}$, respectively) was

used to determine contamination. The spectrophotometer's estimates of $NO_3^-$ were used to quantify contamination. The primary difference in testing the small volume MPS for contamination between sources was the size of the flow-through cell. In this system a 1.1 mL flow-through quartz cuvette was used. In addition to testing cross contamination when rinsing with sample volume from the current source, a DI rinse was tested to see if it resulted in decreased contamination by residuals.

For testing pump timings, external tubes of various lengths were cut and fitted to the inlet port of the system. The length of

time required to pump from the source to fill the cuvette and the time required to purge the line were recorded. Pump times for each tube length were measured twice with use of a stopwatch and average pump time was recorded.

### 2.3.1 Cross contamination results

The cross contamination trial results are summarized in Table 1. The first two columns report p-values testing for significant difference ($\alpha=0.05$) between spectrophotometric estimates of $NO_3^-$ concentration in the source (10 repeated measurements)

and subsequent 10 measurements after pumping sample volume from the alternate $NO_3^-$ source between each measurement (i.e. if p>0.05, there was no statistical difference between initial and subsequent measurements for $NO_3^-$, implying negligible cross contamination). The results show that without a DI rinse and when measuring high concentration samples after low concentration samples the cuvette must be rinsed with the current sample by at least four times $V_{cuvette}$ to make cuvette contamination by previous sample negligible. With rinsing only two times $V_{cuvette}$, concentrations were underestimated at

about 13.25 mg $NO_3$-N L$^{-1}$, lower than the $[NO_3]_{high}$ = 14.72 mg $NO_3$-N L$^{-1}$ reference. Cross contamination was divided by 10 by rinsing with four times $V_{cuvette}$, but was still measurable and significant.





For the low concentration samples measured after high concentrations, rinsing 10 times the $V_{cuvette}$ could not fully eliminate cross contamination as it still appeared to be significant, yielding readings around 0.14 mg $NO_3$-N $L^{-1}$ instead of the initial 0.06 mg $NO_3$-N $L^{-1}$. Adding a DI rinse appeared to eliminate cross contamination for $[NO_3]_{low}$ by rinsing with ten times $V_{cuvette}$ (<0.01 mg/L difference), but the same treatment significantly diluted $[NO_3]_{high}$.

These results suggest that there is no one solution to fully eliminate cross contamination when consecutive samples differ drastically in solute concentration. These results were obtained for extreme changes of conditions where consecutive concentrations were roughly 150-fold different. For applications where this ratio may be common between consecutive samples, rinsing with >10 times $V_{cuvette}$ is recommended and values for lower concentration should be taken with caution. For most applications, however, such ratios between consecutive concentrations are unlikely. For ratios of 50, 10 and 5 (i.e 0.50

to 25.0, 0.5 to 5.0 and 0.5 to 2.5 mg $NO_3$-N $L^{-1}$), the differences in mean concentrations reported in Table 1 would be divided by a factor of 3, 10 and 30, respectively. Rinsing with four times $V_{cuvette}$ would reduce the absolute concentration difference from 0.19 mg $NO_3$-N $L^{-1}$ in the worst-case scenario to 0.06, 0.02 and less than 0.01 mg $NO_3$-N $L^{-1}$, respectively, although the concentrations may still be significantly over- or underestimated. These values are within the 5% measurement uncertainties often found to be acceptable from analytical instruments.

**2.3.2 Source contamination**

Source contamination (or cross contamination of the source) testing results are shown in Fig. 2. As expected, $[NO_3]_{high}$ and $[NO_3]_{low}$ became diluted and concentrated, respectively, as residual volumes from the previous samples contaminated the sources. This source contamination increased with increasing difference in initial concentrations (Fig. 2 and Table 3). Over 40 purges back to the source, the concentration and dilution effects were approximately linear and regression lines were fitted

to the data to calculate effective residual volumes ($V_{res}$). The 95% confidence interval for the slope and intercept of the regression lines were used to calculate the uncertainty on the calculated residual volumes. Standard errors of the regression residuals were twice those of the reference measurements. This indicates additional source(s) of random error, which could include some variability in the residual volumes and/or non-uniform mixing of the sources between consecutive samples. Estimates of $V_{res}$ were calculated to vary between 0.28 and 0.46 mL (Table 2). In trial (a) and (c), $V_{res}$ were statistically

different when calculated using $[NO_3]_{high}$ and $[NO_3]_{low}$, and also differed between trials (a) and (b) for $[NO_3]_{low}$. Practically, our results show that when using the 'purge back to source' configuration (Fig. 1(b)) less than 0.5 mL of water (5% of sample volume pumped) from the previous sample may contaminate the source measured. This is comparable to previous analysis of the large volume MPS showing $V_{res}$ of 4% sample volume pumped (Birgand et al., 2016). This volume may correspond to droplets left in the 0.9 mm ID tubing.

Although this residual volume is significant, its effect depends on the application. When alternating between 3 and 5 mg $NO_3$-N $L^{-1}$ of $NO_3^-$ sources 50 L in volume, it would take 500 purges to source before a 0.01 mg $NO_3$-N $L^{-1}$ change in the concentration would be detectable as a result of residuals purged to the alternate source. In the same situation with a 0.5 L source, only 5 purges would induce the same level of change. These results suggest that the 'purge to source configuration'



should be used in short lived experiments and only in applications with a high source volume (>10 L), which would keep this artifact undetectable. For small-volume applications (e.g. porewater sampling) source contamination may be significant and the 'purge to waste' routine is recommended, rather than 'purge back to source'.

### 2.3.3 Pump time requirement results

5  Times required for a single sampling sequence (pumping and purging from source) for variable tube lengths was not linear with tubing length (data not shown), and best described using the following equation: $Time = 0.0001 \times length^2 + 0.0662 \times length + 79.4$, where *time* corresponds to the cumulative time in s required to pump and purge from the source to the cuvette, and *length* is the tube length from the source to the multiplexer in cm. From this equation, the time for a sampling sequence between consecutive sources was calculated with 60 s added to account for cuvette rinsing for the spectrophotometer measurement time. For applications with sources up to 1, 4, and 9 m away from the MPS, time for one sample sequence would be 147, 182, and 280 s, respectively. The results suggest that roughly 30 min resolution water quality data can be obtained for up to 10 sources with sources up to 4 m away. The resolution would fall to 47 min for the same number of sources up to 9 m away.

## 3 Frequent porewater sampling in a woodchip bioreactor

Woodchip bioreactors, also called denitrification beds, are a popular agricultural conservation practice for the treatment of $NO_3^-$ in subsurface drainage. These anaerobic systems provide woodchips as a carbon source and promote denitrification to remove $NO_3^-$ from the aquatic environment. Soil pits typically ~1 m deep, ~5 m wide and ~25 m long are filled with woodchips though which drainage water percolates. Over twenty years of research on woodchip bioreactors has shown their potential to reduce $NO_3^-$ concentrations, but has also shown troubling variability in reported treatment efficacy (e.g. 81% to 3%; David et.

al 2016), or bioreactor volumetric removal rates (e.g., 2 to 22 g N $m^{-3}$ $d^{-1}$, reviewed by Schipper et. al, 2010; 0.42 to 7.76 g N $m^{-3}$ $d^{-1}$, Chistianson et. al, 2012). Variation in treatment performance has been attributed to various factors, including hydraulic residence time (HRT), temperature, influent concentration, and age (Addy et al., 2016; Hoover et al., 2016). An important part of the reported uncertainty and variability seems to be associated with measurement methods ill-suited to quantify $NO_3^-$ fluxes into, within, and leaving bioreactors. Woodchip bioreactors are mostly treated as 'black box' systems.

There is evidence that internal hydraulic flowpaths and short-circuiting may result in overall treatment inefficiencies, although this has been inferred indirectly in field and lab experiments using tracers (Chistianson et al., 2013; Cameron & Schipper, 2011; Hoover et al., 2016). We report results from a preliminary experiment showing that the small volume MPS can help improve understanding of bioreactors as 'black box' systems.

### 3.1 Materials and methods

Nitrate dynamics were studied in a (1.20 x 1.20 x 0.45 m deep) lab bioreactor at North Carolina State University using the small volume MPS (Fig. S2 and S3). This bioreactor is a lab-scale model of field bioreactors where inlet and outlet manifolds installed at opposite corners create diagonal flow from the inlet at the top to the outlet at the bottom (Fig. S2). Eight sampling



wells within the bioreactor woodchip media were monitored for $NO_3^-$ concentrations using the small volume MPS. Sample wells were placed at shallow (S) or deep (D) zones (20.9 and 41.9 cm depth, respectively) and at 55.9 (In) and 100.2 (Out) cm from the inlet along two longitudinal transects. Transects were located along the centerline (Mid) and 21.6 (Side) cm from left sidewall (Fig. S2). Wells were made of stainless steel tubing (0.32 cm O.D.) with vertical slits cut at the tip to draw water into

the well. Wells tips were covered with fine plankton netting (60 μm mesh) to prevent clogging of MPS tubing. Well names were assigned using the previous codes (e.g. S.In.Mid in the shallow well located at shallow 20.9 cm depth, 55.9 cm from inlet, and along the mid or centerline).

For these experiments a dual sampling/analyzing system was used. The small volume MPS did not directly sample water from each well. Instead, to obtain synchronized $NO_3^-$ concentrations in the bioreactor, all wells were sampled simultaneously using

an 8 channel ISMATEC peristaltic pump triggered by the MPS Arduino board. The 8 channel pump simultaneously delivered sample volumes from each well to an intermediate manifold of eight 40 mL syringes.  The small volume MPS was then used to sequentially pump water to the cuvette for analysis by the spectrophotometer. The probe was calibrated for $NO_3^-$ using PLSR techniques described previously (Etheridge et al., 2014).

### 3.2 Small volume MPS reveals short-circuiting inside a woodchip bioreactor

In the first MPS application, we conducted a 76 h constant $NO_3^-$ injection experiment from April 30 – May 4 2015 in a saturated bioreactor receiving 60 L $h^{-1}$ tap water flow for a theoretical HRT of ~5.7 h. Nitrate from a concentrated $KNO_3$ stock solution was injected using a precision piston pump (Fluid Metering Inc. Model QBG, 1.2 g $L^{-1}$) for a target inflow $NO_3^-$ concentration of 20 mg $NO_3$-N $L^{-1}$. The eight wells were sampled and analyzed every 40 min with the system described in Sec. 3.1. Ports 9 and 10 of the 12 port valve directly sampled the inlet and outlet weirs after consecutively analyzing sample volumes from each

of the eight wells. In the first experiment, $KNO_3$ injection began at 8:10 AM on 5/1 (14.2 h after MPS monitoring began, Fig. 3). From 14.2 to 22.4 h $NO_3^-$ concentrations were stable around 18.3 mg $NO_3$-N/L. After this period inlet concentrations rose until 37 h and was variable over the 76 h injection. Inlet $NO_3^-$ concentration ranged from 18.2 - 20.6 mg $NO_3$-N $L^{-1}$ and varied by 12% of the target inlet concentration. Variability in inlet $NO_3^-$ concentration was higher than anticipated due to degassing in the $KNO_3$ solution tank, with accumulation of air bubbles partially restricted flow though the piston pump.

$NO_3^-$ concentrations at the inlet rose quickly to 18.4 mg $NO_3$-N $L^{-1}$ within 47 min of starting the $KNO_3$ injection (Fig. 3). Inlet $NO_3^-$ concentration likely passed 18 mg $NO_3$-N $L^{-1}$ sooner, but the sampling time resolution of 47 min was insufficient to capture the exact time of the arrival.  Three of the four wells nearest the inlet (S.In.Mid, D.In.Mid, and D.In.Side) showed nearly identical increases in $NO_3^-$, with concentrations at these wells passing 18 mg $NO_3$-N $L^{-1}$ within 4.5 – 8.2 h of the injection. The shallow inlet well located along the side wall (S.In.Side) showed a noticeable lag, with $NO_3^-$ staying below 18

30   mg $NO_3$-N $L^{-1}$ until 19.7 h after the injection began. Nitrate concentrations at this well were significantly lower over the entire injection, relative to other wells near the inlet. All four inlet wells showed increases and decreases in $NO_3^-$ concentrations that corresponded to changes in concentration at the inlet, although variability at S.In.Side was much higher than the other three inlet wells. Nitrate concentrations in deep wells near the inlet closely followed one another during each sample interval, with





an average difference of only 0.13 mg $NO_3$-N $L^{-1}$. Shallow wells near the inlet varied greatly with an average difference of 2.04 mg $NO_3$-N $L^{-1}$.

Four hours after injection $NO_3^-$ was detected at the deep middle well near the outlet (D.Out.Mid), but outlet $NO_3^-$ concentrations did not become stable until about 25 hours, or 5 times the theoretical HT of 5.7 hours. The very long lag for stabilization of $NO_3^-$ concentrations at the outlet weir was likely due to the variability of injected $NO_3^-$ to move through the bioreactor, as apparent from the 16.9, 17.5, and 21.2 h taken for  S.Out.Side, S.Out.Mid, and D.Out.Mid to reach 18 mg $NO_3$-N $L^{-1}$.

Nitrate concentrations were significantly greater in deep wells relative to shallow wells at side wells near the inlet and at middle wells near the outlet. Shallow wells had higher $NO_3^-$ concentration at side outlet wells and middle inlet wells. Variance in $NO_3^-$ concentration was higher in shallow wells than deep wells at all four well pairs, illustrating that water moved quickly diagonally though the bioreactor and short-circuited most of the bioreactor volume. Lag and lower $NO_3^-$ concentrations observed in the shallow wells (e.g., S.In.Side, S.Out.Side, Fig. 4) suggest that areas further away from the direct diagonal flowpath may have slower hydraulic exchange and higher HRT (e.g., S.In.Side ) and act as 'dead zones'.

To our knowledge, short-circuiting and 'dead zones' had never been observed directly but only inferred though high dispersion indices (MDI) shown in field reactors (Chistianson et al.; 2013) and in lab reactors (Hoover et al., 2016). Delayed time of arrival for $NO_3^-$ at wells farther from the shortest flowpath indicate hydraulic inefficiencies, however several shallow wells also showed cyclic patterns in $NO_3^-$ concentrations. Microbial and gas clogging has been documented and can be caused by creation of biofilm pore walls by microbial cells or fungi (Oberdorfer & Peterson, 1985; Okubo & Matsumoto, 1979; Vandivivere et al., 1995) or by extracellular polymers (Shaw et al., 1985; Vandevivere & Baveye, 1992). Gas clogging due to the accumulation of microbial-produced gas bubbles and influencing hydraulic conductivity in peat (Kellner et al., 2004; Beckwith & Baird, 2001) seems to explain some observed transient low conductivity (Kellner et al., 2004). These changes in hydraulic conductivity could explain the $NO_3^-$ concentration fluctuations of the S.Out.Side, S.Out.Mid, and S.In.Side wells, which exhibited regular decreases and increases of $NO_3^-$ concentrations

### 3.3 Internal nitrate removal kinetics

In a second set of bioreactor experiments, $NO_3^-$ reduction kinetics were measured at each well. The lab bioreactor was fully drained for a period of 24 hours. Following this drain event, the woodchips were re-saturated with the same $KNO_3$ spiked tap water made by the same method described in Sec. 3.2. Inflow $NO_3^-$ concentration was 14 mg $NO_3$-N $L^{-1}$ at 108 L $h^{-1}$. After pumping 4.5 pore volumes though the reactor, flow was stopped and the outlet weir raised to prevent any outflow. Each well was sampled every 3 h over 24 h during five trials from March 26 – April 12 using the MPS and 8 channel pump. Times series of $NO_3^-$ concentrations at each well were fitted to zero order kinetics models (Eq. 1.1). Time series were fitted using the nls() function in R which calculates least-squares estimates for model parameters. In fitting the zero order model, only the concentrations after flow was stopped and above 2 mg $NO_3$-N $L^{-1}$ were used.  Other than both shallow outlet wells, $NO_3^-$ concentrations at each well peaked at 12.5-13.5 mg $NO_3$-N $L^{-1}$ after flow was stopped. Both shallow outlet wells had much





lower peaks of 9.5-10.5 mg NO$_3$-N L$^{-1}$, consistent with observations in the first experiment that wells in this zone were slow to receive new water.

Over the five trials zero order NO$_3^-$ removal rates ranged from 0.13 to 0.54 mg NO$_3$-N L$^{-1}$ h$^{-1}$ (Table 3). For this experimental bioreactor with a measured porosity of 0.59, this equates to a range of 1.84 – 7.65 g N m$^{-3}$ d$^{-1}$ (m$^3$ of reactor volume). Having

access to eight points provided a range of removal rates and illustrated variability within the bioreactor. Spatial variability in the volumetric NO$_3^-$ removal rates approached those reported across many field bioreactors (reviewed by Schipper et al., 2010). While there was high variability in NO$_3^-$ removal rates within trials, there was no significant difference in mean values between wells when considering all five trials. The five lowest removal rates measured (Table 3) were in shallow outlet wells (observed dead zones), and indicate that there may be a causality between NO$_3^-$ removal rates and hydraulic inefficiencies or clogging.

**4 High frequency measurements in *in situ* stream mesocosms**

Among the many methods used to measure NO$_3^-$ removal kinetics in streams, the *in situ* mesocosm method is attractive as it involves minimal disturbance of the sediment, is effective for investigating spatial variability under field condtions, and can be performed at different times of the year (reviewed by Birgand et al., 2007). *In situ* mesocosms consist of open bottom containers inserted into the sediment which isolate water inside the mesocosm from the surrounding stream, making it possible

to estimate process kinetics from changes in nutrient concentrations over time. Water recirculation is often applied to mimic ambient stream velocity.

The standard method consists in manual sample withdrawal several times (most often 4-7 times; reviewed by Birgand et al., 2007) over the duration of the experiment (typically <48 h). In laboratory mesocosms the volume of the water withdrawn is a minor issue as one can account for the mass of nutrients removed (e.g., Birgand et al., 2016; Messer et al., 2017). For *in situ*

mesocosms water withdrawal is more significant as this water will, over time, be replaced by water upwelling from the sediment porewater. This discourages taking more samples during the experiment to reduce uncertainty in kinetics estimates. The inherent conflict between kinetics uncertainties and porewater interference in *in situ* stream mesocosms can be solved with the small volume MPS used in the 'back to source' configuration (Fig 1(b)).

**4.1 Materials and methods**

To characterize NO$_3^-$ removal kinetics of a stream prior to its restoration (Claridge Nursery, Goldsboro, NC; 35.4° N, 78.0° W), the small volume MPS was used with *in situ* mesocosms in two distinct sections of the reach to be restored. The upper reach of the ditch (1.6 km) has submerged vegetation and a thick muck sediment (20-30 cm depth) with high organic content (2-26% organic matter). The lower reach (0.6 km) has no instream vegetation and sandy sediment with low organic content (0.1-9.0 % organic matter). The upper and lower reaches are referred to as Muck and Sand, respectively.

In eleven 24 h experimental trials from August 2014 – March 2015, four mesocosms made of open bottomed barrels were gently inserted into the sediment down to approximately 10 cm within a 1 m radius of each other (Fig. S4). Each 57 cm diameter barrel inserted into the sediment are referred to as Sediment and covered 0.26 m$^2$ of stream bottom. A fifth, closed



bottom barrel (referred to as Control) placed in the stream for temperature adjustment and containing only stream water served as a control, representing  $NO_3^-$ removal processes occurring in the water column. Water depth in Sediment mesocosms was measured manually. Recirculating DC pumps (3 L min$^{-1}$) were placed on the sidewall of the barrel (Fig. S5) and their flow was adjusted to mimic ambient stream velocity (0.02 – 0.10 m s$^{-1}$). Sample lines were placed near the recirculating pumps to

obtain a well-mixed sample volume. Mesocosms were removed at the end of each experiment to allow hydrologic connection with the stream between trials. Four 24 h trials were completed at each Muck site and three trials performed at the downstream Sand site across four seasons.

At the beginning of each trial a 2 L solution of $KNO_3$ was added to each mesocosm to reach an initial $NO_3^-$ concentration of 5-6 mg $NO_3$-N L$^{-1}$. The overlying water was gently stirred and left to equilibrate.  The addition of the $KNO_3$ solution generating

extra head inside the mesocosm helped prevent upwelling from groundwater (Solder et al., 2015) which might have otherwise occurred. Each sample volume pumped by the MPS temporarily withdrew ~25 mL of sample over <4 min, corresponding to <0.1 mm drop in head, which we assumed was not high or long enough to generate significant upwelling.

Mesocosms were sampled every 36 min for 24 h using the small volume MPS. The measurement cuvette (1.1 mL, 4 mm path length) was rinsed with >10 times $V_{res}$ to prevent cross contamination. Zero order, first order, and efficiency loss (EL) kinetics

models accounting for water depth (D) (Eq. 1.1-1.3) were fitted to the $NO_3^-$ concentration time series (~40-50 data points per mesocosm per trial) to 1) compare $NO_3^-$ removal rates between sites and across seasons and 2) determine which model best predicted the observed data.

$$C_t = C_0 - \frac{\rho_{ZO}}{D} * t \tag{1}$$

$$C_t = C_0 * e^{-\left(\frac{\rho_{Fo}}{D}*t\right)} \tag{2}$$

$$C_t = \left(\frac{\rho_{EL}^{(\alpha-1)}}{D} * t + C_0^{(1-\alpha)}\right)^{\left(\frac{1}{1-\alpha}\right)} \tag{3}$$

Methods of model fitting and evaluation were the same as those presented in Sec. 3.3 (Messer et al.,2017; Birgand et al.,2016). The *nls* package in R Studio was used to fit observed data to a model predicting $NO_3^-$ concentration at time t, $C_t$, from estimates of initial concentration, $C_0$, depth-compensated removal constants ($\rho_{ZO}$, $\rho_{FO}$, $\rho_{EL}$),  and the efficiency loss constant $\alpha$. Commonly reported rate constant, k, was calculated by dividing $\rho$ coefficients by depth, D.

**4.2 Nitrate uptake kinetics**

There was little preference between zero and first order models in the time series of decreasing $NO_3^-$ concentrations (Table 4), with $R^2$ values for both models typically >0.95. Residuals of both models were nearly identical in most cases and differences in model $R^2$ were generally <0.01. All time series of decreasing $NO_3^-$ concentrations from Sand trials showed better fit for zero order kinetics. Higher $R^2$ and improved residuals for a first order model, relative to a zero order model, were seen in trials with

greater decreases in $NO_3^-$ concentrations over the experiment. This was the case in Mar-18 and Aug-28 trials where net $NO_3^-$



reduction in several mesocosms approached 2-3 mg $NO_3$-N $L^{-1}$ over 24 h. The poorest model fitting occurred when fitting models to time series with little to no reduction in $NO_3^-$ concentrations, when total variability of $NO_3^-$ was close to the precision of the spectrophotometer or transient, short-term changes in $NO_3^-$ were large relative to net reduction. The EL model was not useful in these short-term experiments.

These results are somewhat contradictory to results obtained in lab wetland mesocosms of similar size where data were not well-fitted by a zero order model and where the EL model was the best (Birgand et al., 2016). It is possible that the experiments were not long enough for $NO_3^-$ concentrations to decrease enough for differences to appear among models, or for first order and EL kinetics to be apparent. In these *in situ* experiments, the results suggest that $NO_3^-$ removal kinetics at the diurnal scale in nutrient-rich streams were adequately predicted by zero order models. In Muck trials, ρ values ranged from 10 – 580 mg

$NO_3$-N $m^{-2}$ $d^{-1}$ (Table 4). Removal rates in Sand mesocosms were much lower, ranging from 10 – 240 mg $NO_3$-N $m^{-2}$ $d^{-1}$. The results show that for $NO_3^-$ removal rates less than 300 mg N $m^{-2}$ $L^{-1}$, the zero order rate model was sufficient to fit the data, but the first order model appeared better for higher $NO_3^-$ removal rates generally found in agricultural streams (350 and 1,250 mg N $m^{-2}$ $d^{-1}$; Birgand et al., 2007).

### 4.3 Seasonal and spatial variability

In Muck trials $NO_3^-$ removal rates followed a predictable seasonal pattern with removal rates highest in the month of August, decreasing in October and November trials, lowest during the winter months, and increasing again during March. This is consistent with observations that the rates of microbial processes increase with temperature. Nitrate removal rates during the Aug-18 trial (25 °C) were 4-14 times greater than those during Feb-3 (8 °C) at the same site. The opposite trend was seen among Sand trials. The Dec-17 trial with the coldest temperatures among Sand trials (10 °C) showed the highest $NO_3^-$ removal

rates. A seasonal influx of available carbon was likely the cause of this trend. Accumulated leaf packs at the Sand site, included in the Dec-17 mesocosms, provided available carbon and substrate for denitrification to occur. Nitrate removal rates during this trial were even higher than rates in Muck trials with similar temperature (Jan-29 and Feb-3).

The *in situ* mesocosm method revealed high variability in $NO_3^-$ removal rates within trials, even when mesocosms were within a 1 m radius. High variability in Muck trials in the month of March was caused by the presence of emergent vegetation along

the stream bank (Fig. S6). In Mar-11 and Mar-18 trials, mesocosms placed in this near-bank vegetated zone showed removal rates 48-81% and 74-240% higher than those in mesocosms placed in the unvegetated stream center, respectively. In the Jan-29 trial, a single mesocosm showed removal rates 300-2500% higher than the other Sediment mesocosms. Muck trials had a higher degree of within-trial variability relative to Sand, indicating that sediment $NO_3^-$ removal potential at the Sand site was more homogenous.

### 30 5 Conclusions

The first report of the MPS illustrated the ability of this technology to increase temporal and spatial resolution of water quality data (Birgand et al., 2016). The small volume MPS increases the number of potential applications for this method by



significantly decreasing sample volume. The small volume MPS minimized the volume of the flow-through measurement cell (1.1 mL quartz cuvette), contrasting with the 40 mL flow-through cell used previously. In order to prevent cross contamination of samples caused by pumping from different sources, an adequate pre-measurement rinse with the current sample volume must be used. For the larger volume MPS this would require pumping roughly 36 times more volume past the flow-through

cell for adequate rinsing. In the stream mesocosms, removing 25 mL of sample for <4 min to measure absorbance resulted in temporary head drop <0.1 mm and minimized sample withdrawal to allow 40-50 $NO_3^-$ measurements to be made over 24 h without significantly affecting mesocosm hydraulics.

The small volume MPS allows continuous multi-point sampling to be extended to applications where sample volume is limited or must be minimized. The most obvious application is for porewater sampling, where sample withdrawal rate should not

exceed the rate at which water moves though the medium. Total volume extracted must also be small to avoid significantly affecting the observed environment. Drainable porosity in soils is much less than the total volume of soil, leading to a zone of influence which size depends on the total sample volume extracted and the drainable porosity. For example, sampling every 1 or 6 h over 24 h from a soil with a 0.1 drainable porosity, this zone of influence (assuming 15 mL required sample volume for the small volume MPS) would be 3600 and 600 $cm^3$, respectively. Assuming this zone is spherical, it would have a diameter

of 19 and 10 cm, which is not insignificant. Sampling interval should be short enough to adequately capture temporal variation while avoiding excessive water withdrawal to avoid interdependence between sampling points that are hydraulically connected. When removal kinetics were measured in the lab bioreactor during stopped flow, a sample interval of 3 h was used over 24 h. With an observed drainable porosity of 0.58, the zone of influence at each of the eight points was 7.3 cm and accounted for 0.4% of water in the bioreactor. Sampling wells were no closer than 21 cm, so an assumption of independence

between sampling points was reasonable.

The small volume MPS was able to observe for the first time short-circuiting in a woodchip bioreactor which had been surmised in the literature although never fully shown. We were also able to measure $NO_3^-$ removal rates at multiple locations within the bioreactor. The application of the MPS in stream mesocosms opened the possibility to measure $NO_3^-$ removal kinetics *in situ* in replicated experiments by eliminating potential contamination by groundwater. The small-volume MPS has high potential

for providing quality data sets for improving new and existing solute transport models for saturated or partially saturated soils and opens the possibility to be used in replicated experiments.

**5.1 System limitations and recommendations**

Due to the small diameter of the fittings and tubing in this system, several pumping concerns are magnified compared to the large volume MPS. Water micro-droplets or residuals in the tubing lines are more susceptible to freezing and caution should

be used when deploying this system under freezing conditions. Valves and tubing are also more susceptible to clogging. In both applications a plankton net fabric (60 μm mesh) was used and no clogging occurred. The small sample volume potentially limits the number of sondes and sensors to which the MPS can be coupled. The design of the s::can spectrophotometer allows





for such a small volume flow-through measurement cell to be used, while other available water quality sensors typically require much larger sample volumes for which this system might not be well-suited.

*Data availability:* All data presented in this paper in Sec. 2.3.1, 2.3.2, 3.1, 3.2, 4.2, and 4.3 are provided in .csv files on an online GitHub repository at https://github.com/bmmaxwel/Maxwell.et.al.2018_SmallVolumeMPS. The online repository also includes the R code (R Studio Ver.1.1.442 or greater) used to generate plots and calculate coefficients for $NO_3^-$ removal kinetic equations.

*Competing interests:* The authors declare that they have no conflict of interest.

*Author contributions:* Direction of research, funding, and initial development of the small volume MPS was provided by François Birgand. Early design and prototyping of the small volume MPS was provided by Brad Smith and Kyle Aveni-Deforge, and development of protocols for testing cross contamination were contributed to by Kyle Aveni-Deforge.

*Acknowledgements:* The authors would like to acknowledge the NCSU BAE Environmental Analysis Lab for water chemistry analysis. Stream mesocosm research was funded by the NCDOT under grant agreement 2013-37, and bioreactor porewater sampling research was funded in part by NIFA USDA grant 2016-67019-25279 and USDA multistate project S1063.

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





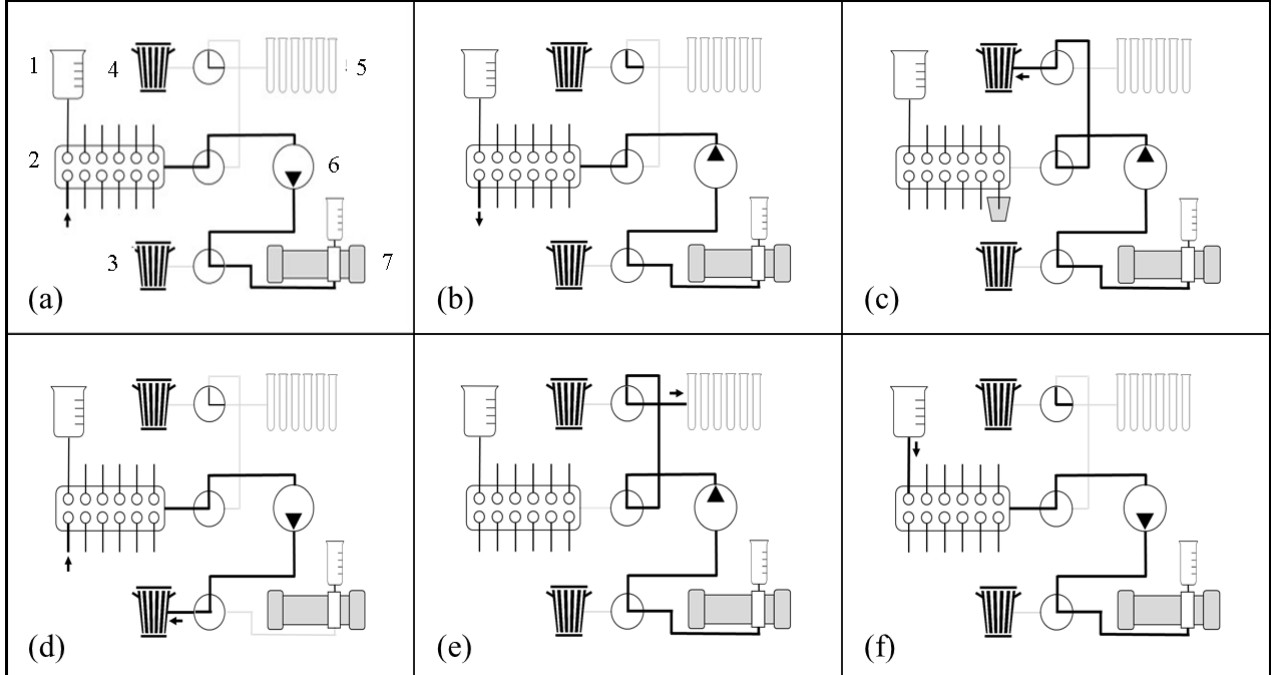

**Figure 1 – Graphical depiction of small volume multiplexed pumping system (MPS) configurations which include 1) DI water source, 2) 12 port intake valve, 3 & 4) waste or air purge, 5) fractional volume collector (FVC), 6) bidirectional peristaltic pump, and 7) spectrophotometer with cuvette housing. An Arduino microcontroller actuates a series of 3 way valves to move between separate configurations described in Sec. 2.2.1 – 2.2.5.**





**Figure 2 - Changes in NO₃⁻ concentrations in low ([NO₃]low) and high ([NO₃]high) concentration sources (0.5 L) resulting from residual volumes purged to the alternate source. Greater concentration or dilution effects were seen with increasing differences in initial NO₃⁻ concentrations of the sources during Trials (a) – (d).**




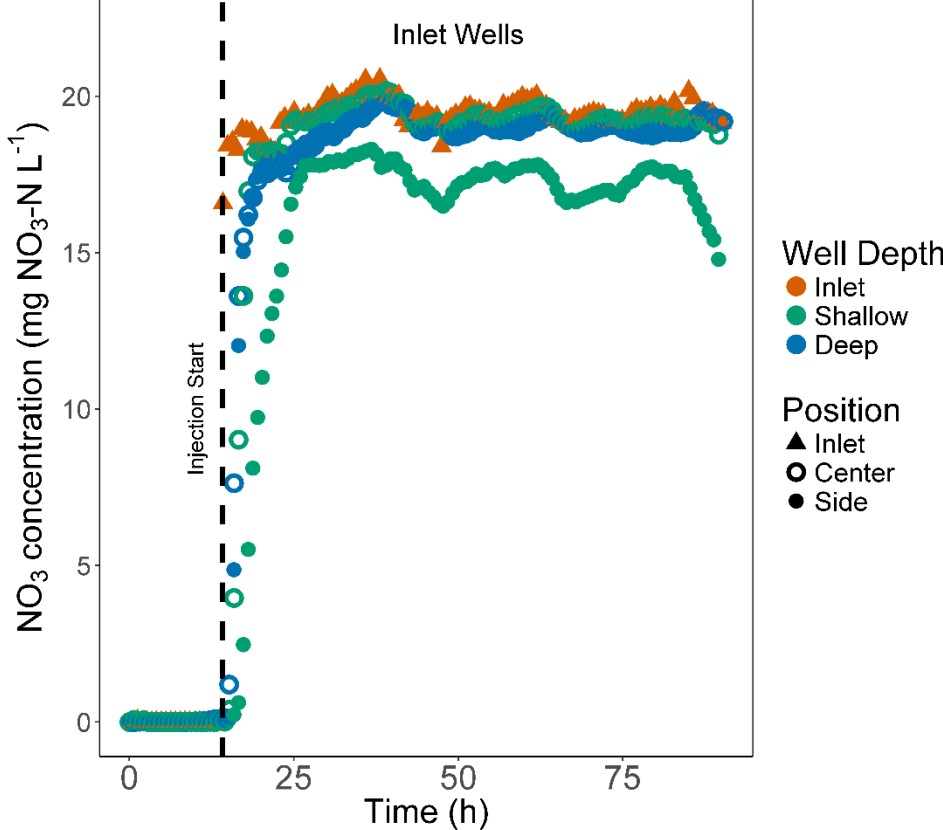

**Figure 3 – Nitrate concentration at inlet wells before and during the KNO₃ injection. Color indicates bioreactor inlet and depth of sampling wells, shape indicates the position of the well transect (centerline or near side wall).**



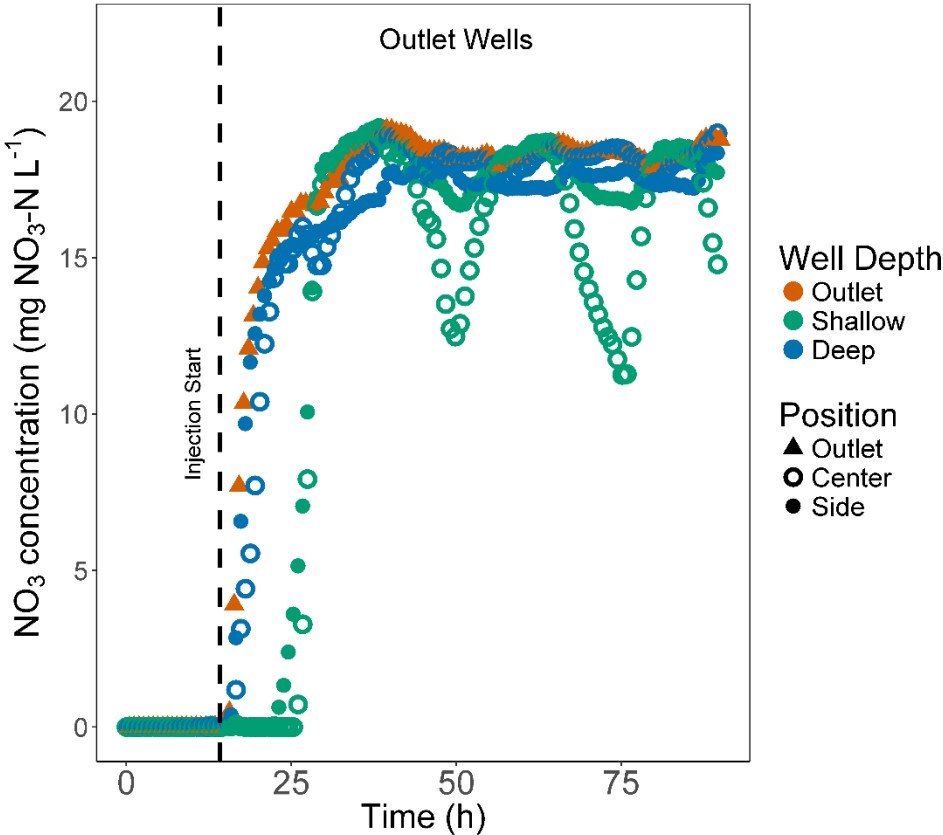

**Figure 4 – Nitrate concentration at outlet wells before and during the KNO₃ injection. Color indicates bioreactor outlet and depth of sampling wells, shape indicates the position of the well transect (centerline or near side wall).**

**Table 1 - Results of testing cross contamination of cuvette. Treatment indicates the volume of sample pumped from the current source for cuvette rinsing relative to cuvette volume (V_cuvette), and whether or not a prior DI rinse was used. *** indicates that the protocol used resulted in no significant difference between the initial NO₃⁻ concentration and subsequent measurements (α=0.05).**

| Trial # | Treatment | Initial concentrations (mg NO₃-N L⁻¹) | | p-value for difference in means of [NO₃]High | p-value for difference in means of [NO₃]Low | 95% C.I. for difference in means, [NO₃]High (mg NO₃-N L⁻¹) | 95% C.I. for difference in means, low NO₃⁻ source (mg NO₃-N L⁻¹) |
|---|---|---|---|---|---|---|---|
| | | [NO₃]High | [NO₃]Low | | | | |
| 1 | 2x V_cuvette | 14.71 ± 0.02 | 0.12 ±0.01 | <0.0001 | <0.0002 | (-1.59, -1.37) | (1.29, 2.10) |
| 2 | 4x V_cuvette | 14.71 ± 0.01 | 0.09 ±0.01 | <0.0001 | <0.0001 | (-0.19,-0.14) | (0.15,0.18) |



| 3 | 10x $V_{cuvette}$ | 14.75 ± 0.01 | 0.06 ±0.01 | 0.236*** | <0.0001 | (-0.03,0.01) | (0.06,0.09) |
| 4 | DI rinse, 2x $V_{cuvette}$, | 14.74 ± 0.01 | 0.08 ±0.01 | <0.0001 | <0.0001 | (-1.74,-1.41) | (0.28,0.35) |
| 5 | DI rinse, 4x $V_{cuvette}$ | 14.74 ± 0.01 | 0.11 ±0.02 | <0.0001 | <0.0001 | (-0.11,-0.07) | (0.04,0.08) |
| 6 | DI rinse, 10x $_{cuvette}$ | 15.14 ± 0.01 | 0.09 ±0.02 | 0.003 | 0.615*** | (-0.06,-0.02) | (-0.02,0.04) |

Table 2: Initial $NO_3^-$ concentrations, calculated effective residual volumes (Vres) and 95% confidence interval (C.I.) from source contamination Trials (a) – (d).

| | $[NO_3]_{Low}$ | | | $[NO_3]_{High}$ | | |
|---|---|---|---|---|---|---|
| Trial | Initial concentration (mg $NO_3$-N $L^{-1}$) | $V_{res}$ (mL) | 95 % C.I. (mL) | Initial concentration (mg $NO_3$-N $L^{-1}$) | $V_{res}$ (mL) | 95 % C.I. (mL) |
| (a) | 0.71 | 0.41 | (0.39, 0.43) | 15.3 | 0.32 | (0.29, 0.35) |
| (b) | 0.71 | 0.35 | (0.32, 0.38) | 8.63 | 0.39 | (0.34, 0.43) |
| (c) | 0.74 | 0.46 | (0.38, 0.55) | 4.80 | 0.28 | (0.21, 0.36) |
| (d) | 0.78 | 0.30 | (0.06, 0.52) | 2.52 | 0.39 | (0.15, 0.60) |

Table 3 – Zero order $NO_3^-$ removal rates, *k*, at eight well locations during five experimental trials. Zero order kinetics model was fitted to time series of $NO_3^-$ concentrations after flow was stopped to the lab bioreactor.

| Well Location | Zero order, *k* (mg $L^{-1}h^{-1}$) | $R^2$ | RMSEP (mg $L^{-1}h^{-1}$) |
|---|---|---|---|
| D.In.Mid | 0.25 – 0.43 | 0.96 – 0.99 | 0.002 – 0.003 |
| D.In.Side | 0.27 – 0.43 | 0.94 – 0.99 | 0.002 – 0.005 |



| | | | |
|---|---|---|---|
| D.Out.Mid | 0.29 – 0.38 | 0.84 – 0.99 | 0.002 – 0.010 |
| D.Out.Side | 0.30 – 0.38 | 0.93 – 0.99 | 0.001 – 0.008 |
| S.In.Mid | 0.30 – 0.44 | 0.93 – 0.99 | 0.002 – 0.008 |
| S.In.Side | 0.30 – 0.54 | 0.97 – 0.99 | 0.002 – 0.006 |
| S.Out.Mid | 0.15 – 0.34 | 0.88 – 0.98 | 0.003 – 0.006 |
| S.Out.Side | 0.13 – 0.36 | 0.91 – 0.99 | 0.001 - 0.010 |

**Table 4 – Time series of $NO_3^-$ concentrations in Sediment mesocosms were fitted to zero and first order kinetics models. Results indicate that for this short duration experiment (<24 hr) and at the observed range of $NO_3^-$ concentrations (2-6 mg $NO_3$-N $L^{-1}$), removal was described equally well by either model, although first order kinetics better fitted $NO_3^-$ time series during trials with large changes in $NO_3^-$ concentration.**

| Trial Date (Sediment) | Zero-order, $k$ mg $NO_3$-N $m^{-2}$ $d^{-1}$ | $R^2$ | RMSEP mg $NO_3$-N $m^{-2}$ $d^{-1}$ | First-order, $k$ m $d^{-1}$ | $R^2$ | RMSEP m $d^{-1}$ |
|---|---|---|---|---|---|---|
| Aug. 28 (Muck) | 50 <br> 0 - 580 | 0.99 | 30 - 60 | 0.15 - 0.17 | 0.99 | 0.02 - 0.05 |
| Oct. 1 (Muck) | 200 - 250 | 0.95 – 0.98 | 30 - 60 | 0.06 – 0.07 | 0.95 – 0.98 | 0.03 – 0.05 |
| Nov. 6 (Muck) | 140 - 260 | 0.93 – 0.98 | 40 - 70 | 0.04 – 0.07 | 0.93 – 0.98 | 0.03 – 0.06 |
| Nov. 13 (Muck) | 160 - 200 | 0.89 – 0.98 | 20 - 60 | 0.05 – 0.06 | 0.89 – 0.98 | 0.02 – 0.06 |
| Jan. 29 (Muck) | -40 - 180 | 0.00 – 0.93 | 10 - 40 | 0.00 – 0.05 | 0.00 – 0.93 | 0.00 – 0.04 |
| Feb. 3 (Muck) | 40 - 120 | 0.76 – 0.99 | 20 - 30 | 0.01 – 0.04 | 0.76 – 0.99 | 0.02 – 0.03 |
| Mar. 11 (Muck) | 180 - 320 | 0.93 – 0.99 | 20 - 40 | 0.04 – 0.09 | 0.92 – 0.99 | 0.02 – 0.04 |
| Mar 18 (Muck) | 90 - 300 | 0.97 – 0.99 | 20 - 60 | 0.03 – 0.13 | 0.97 – 0.99 | 0.02 – 0.05 |



| | | | | | | |
|---|---|---|---|---|---|---|
| Sep 18 (Sand) | 10 - 110 | 0.10 – 0.86 | 20 - 50 | 0.00 – 0.03 | 0.10 – 0.85 | 0.02 – 0.05 |
| Dec 17 (Sand) | 140 - 240 | 0.97 – 0.98 | 20 - 30 | 0.03 – 0.06 | 0.96 – 0.98 | 0.02 – 0.03 |
| Mar 16 (Sand) | 20 - 50 | 0.63 – 0.84 | 20 - 30 | 0.01 – 0.02 | 0.63 – 0.86 | 0.02 – 0.03 |