# Peer review of "A small volume multiplexed pumping system for automated, high frequency water chemistry measurements in volume-limited applications"

_Hydrology and Earth System Sciences, 2018_

## Referee Comment (RC1) · Anonymous Referee #1 · 10 Jul 2018

Article Summary

The authors describe the use of a novel pumping system that uses bidirectional flow to sample pore water using a spectro-lyser with an in-line cuvette. The paper is well written and detailed, and I commend the authors on their approach that captures both temporal and spatial variability, which is by no means an easy feat. I particularly enjoyed the inclusion of the woodchip bioreactor experiment. Additionally, the authors take good care with their design scheme and descriptions to ensure that a user could set up a similar device. Please see my detailed, mostly minor, comments below, which

I hope the authors find insightful and useful.

I would recommend a minor revision of this manuscript.

Major Comments

I have no major comments for this manuscript.

Minor Comments:

Line-by-Line Comments

Highlights/Abstract

Page 1.10: The authors state that their MPS system was previously reported, but this phrasing seems awkward without direct reference to the citation.

Page 1.20: "The technology is most promising". . . I would mention at what spatial scale (i.e., I presume reach scale or transect style studies) and perhaps temporal scale your technology would be best for.

Main Text

Section 1.1 Page 2.10: "Multiplexing" can mean different things across disciplines. I would just be very clear how you are using that term here.

Section 1.1 Page 2. 15: Do you mean soils or sediments here? Either way, both should be mentioned given your application.

Section 1.1 Page 2.25: If one of the challenges is tube clogging, it would be useful to describe at some point the limitation of the instrument. For example, if the system has high organic matter, would that system be a candidate for your instrument? Are there ways to avoid tube clogging to make it work in that type of system?

Section 2.3 Page 5.5: "Although this is admittedly undesirable. . ." Can you describe in greater detail the solutions to minimize risk here?

[Figure]

Section 3.2 Page 9.15-20: I really enjoyed the discussion of the short circuiting of the reactor.

Figures/Tables

Figure 1: Please also include the lettering system in your figure description.

Figure 3-4: The symbols are slightly difficult to distinguish. Putting the y axis on a log scale might help.

Would it be possible to include a picture of your experimental setup in the SI?

---

## Author Comment (AC1) · 18 Jul 2018

Page 1.10: The authors state that their MPS system was previously reported, but this phrasing seems awkward without direct reference to the citation.

Response : Text has been edited to clarify this comment : "An automated multiplexed pumping system (MPS) for high frequency water chemistry measurements at multiple locations previously showed ability to increase spatial and temporal resolution of data and improve our understanding of biogeochemical processes in aquatic environments

and at the land-water interface." The HESS Manuscript Guideline instructions say to not include references in the Abstract, but this reference can be added if this is recommended by the Referree.

Page 1.20: "The technology is most promising". . . I would mention at what spatial scale (i.e., I presume reach scale or transect style studies) and perhaps temporal scale your technology would be best for.

Response : Edited text : "The technology is most promising at the reach or transect scale for observing pore water solute dynamics over daily time scales, with data intervals <1 h for up to 12 locations."

Main Text Section 1.1 Page 2.10: "Multiplexing" can mean different things across disciplines. I would just be very clear how you are using that term here.

Response : Sentence added for clarification : "The MPS is a 'multiplexed' system in that it delivers sample volumes from separate sources to a single probe used to consecutively observe water chemistry at all sources."

Section 1.1 Page 2. 15: Do you mean soils or sediments here? Either way, both should be mentioned given your application.

Response : Updated to include "sediment"

Section 1.1 Page 2.25: If one of the challenges is tube clogging, it would be useful to describe at some point the limitation of the instrument. For example, if the system has high organic matter, would that system be a candidate for your instrument? Are there ways to avoid tube clogging to make it work in that type of system?

Additions were made to Section 5.1 : "Valves and tubing are also more susceptible to clogging. In both applications a plankton net fabric (60 $\mu$m mesh ) was used and no clogging occurred, even in the case of the woodchip bioreactor application with high dissolved organic matter." In the authors' extensive use of this system in multiple applications, there has not been an issue where clogging occurred as long as plankton

mesh filters were used.

Section 2.3 Page 5.5: "Although this is admittedly undesirable. . ." Can you describe in greater detail the solutions to minimize risk here?

Response : Edited text "Potential solutions for minimizing cross-contamination include a pre-measurement rinse with the current source water or extended purging of the lines with air after each measurement."

Section 3.2 Page 9.15-20: I really enjoyed the discussion of the short circuiting of the reactor.

Figure 1: Please also include the lettering system in your figure description.

Figure caption edited to include letter description. Updated caption : "Graphical depiction of small volume multiplexed pumping system (MPS) configurations which include 1) DI water source, 2) 12 port intake valve, 3 & 4) waste or air purge, 5) fractional volume collector (FVC), 6) bidirectional peristaltic pump, and 7) spectrophotometer with cuvette housing. An Arduino microcontroller actuates a series of 3 way valves to move between separate configurations 1(a) – 1(f) described in Sec. 2.2.1 – 2.2.5."

Figure 3-4: The symbols are slightly difficult to distinguish. Putting the y axis on a log scale might help.

Figures corrected to show nitrate concentrations at each well in separate panels to make points easier to distinguish, as a log scale did not help significantly for this data. See attached figure.

Would it be possible to include a picture of your experimental setup in the SI?

The authors are not clear on which experimental set up photos are being requested. Currently there are photos of the mesocosm experimental set up and the woodchip bioreactor in the SI. Depending on which photos are being requested, will try to include these in the supplemental information.

[Figure]

Please advise where to post the updated manuscript with updated figures and text. Thank you for your comments!

[Figure]

**Fig. 1.**

[Figure]

Figure 4 - Outlet wells

**Fig. 2.**

---

## Referee Comment (RC2) · Anonymous Referee #2 · 19 Jul 2018

In the present manuscript, the authors designed an integrated system for small-volume and high frequency water chemistry measurements. They elaborated the hardware components, the system configurations and the operation notes of the system. Laboratory and field applications of this system were also exhibited to demonstrate the usefulness of the system. The automated feature for sample collection and determination of water chemistry suits very welll in continuous monitoring in a water environment. Therefore, I suggest minor revision for this manuscript.

Major comments: No major comments.

Specific comments: 1.Although the well names (S.In.Mid, S.In.side, etc) used in the manuscript are understandable, but I found it very difficult to follow when I am trying to interpret the data plots. I often need to stop and recall the exact meaning of each name. So I strongly suggest the authors to replace the well names. Maybe S1- S4 for the wells in shallower depth and D1-D4 for the welss in deeper depth.

2.Page 5, Line 27-29. "The results show that without a DI rinse. . . by at least four times Vcuvette to make . . .". Is this correct? Because the p = 0.236 for 10x Vcuvette, which suggest negligible difference after 10 times purge, right?

3.Page 8, Line 13, determination of NO3- with optical methods should be detailed either in the manuscript or in the supporting information.

4.Page 10, Line 33, should be "referred to as Sediment mesocosms"

5.Although it is stated as small-volume sampling, it usually withdrew tens of milliliters of water, which might not be a big problem for overlying water. But the volume is relatively large when this device is applied to extract porewater from porous media, for example in the case of sediments. This could be a limitation for the implementation of the system designed. Also as a suggestion for the authors to improve the system in the future, maybe rhizon in situ samplers (Seeberg-Elverfeldt et al., 2005 Limnol. Oceanogr: Method 3, 2005, 361-371) could be included into the system as the fine pores on the sampler preclude the suction of fine particles (diameter > 0.2 $\mu$m).

6.Many typos such as "PTFE", "through" should be corrected.

---

## Author Comment (AC2) · 20 Jul 2018

Major comments: No major comments.

Discussion paper Specific comments:

1.Although the well names (S.In.Mid, S.In.side, etc) used in the manuscript are under-standable, but I found it very difficult to follow when I am trying to interpret the data plots. I often need to stop and recall the exact meaning of each name. So I strongly suggest the authors to replace the well names. Maybe S1- S4 for the wells in shallower

depth and D1-D4 for the welss in deeper depth.

Response : Corrected wells names to D1-D4 and S1-S4 for shallow and deep wells, as well as edited the references to them in Sec. 3.1 – 3.2

2.Page 5, Line 27-29. "The results show that without a DI rinse. . . by at least four times Vcuvette to make . . .". Is this correct? Because the p = 0.236 for 10x Vcuvette, which suggest negligible difference after 10 times purge, right?

Response : This is correct, and thank you for catching this. This has been corrected in the manuscript. "the cuvette must be rinsed with the current sample by at least ten times Vcuvette to make cuvette contamination by previous sample negligible"

3.Page 8, Line 13, determination of NO3- with optical methods should be detailed either in the manuscript or in the supporting information.

Response : In Sec. 2.3, the following edits were made : "The spectrophotometer's estimates of NO3- based on the absorbance fingerprint (method detailed further in Supporting Information) were used to quantify contamination."

In the Supporting Information, the following paragraph was added to clarify the optical method for estimating nitrate from absorbance.

The s::can spectro::lyser (s::canTM, Vienna, Austria) was used for the optical measurement of NO3- concentrations during the source and cross contamination trials (Sec. 2.3.1 and 2.3.2), as well as the woodchip bioreactor and stream mesocosm applications. The probe's estimates for NO3- were used to determine source/cross contamination due to the probe's low coefficient of variance for NO3- measurements and ability to detect small changes in NO3- concentration. Measurement of NO3- by the spectrophotometer is made possible by the principle of Behr's law and measuring the absorbance of a water volume over the 200 – 750 nm range. The resulting absorbance fingerprint provides absolute absorbance values at each 2.5 nm interval, with NO3- showing an absorbance peak at 200-205 nm. The spectrophotometer was configured for the 4 mm

pathlength cuvette by changing the measurement pathlength of the spectrophotometer, and a baseline using deionized water was set using the manufacturer's specifications. While the spectro::lyser provides estimates of NO3- concentrations based on the manufacturer's global calibration, a local calibration is often recommended, particularly as the presence of organic matter absorbing at UV wavelengths can interfere with measurement of nitrate. During the cross and source contamination, the global calibration of the probe was used. In the bioreactor and stream applications, a local calibration was performed using PLSR methods detailed further in Etheridge et al. (2014). The R package pls was used to construct a model using the 240 raw absorbance values as predictors for the lab samples analyzed for NO3- by the NCSU Environmental Analysis Lab.

4.Page 10, Line 33, should be "referred to as Sediment mesocosms"

Response : Corrected to include "referred to as Sediment mesocosms". Same with "Control mesocosms."

5.Although it is stated as small-volume sampling, it usually withdrew tens of milliliters of water, which might not be a big problem for overlying water. But the volume is relatively large when this device is applied to extract porewater from porous media, for example in the case of sediments. This could be a limitation for the implementation of the system designed. Also as a suggestion for the authors to improve the system in the future, maybe rhizon in situ samplers (Seeberg-Elverfeldt et al., 2005 Limnol. Oceanogr: Method 3, 2005, 361-371) could be included into the system as the fine pores on the sampler preclude the suction of fine particles (diameter > 0.2 $\mu$m).

Response : the following edits were made in Sec. 5.1, detailing the system limitations. "The volume-limited applications presented include sampling an overlying water column and porewater in coarse woodchips. Porewater sampling in fine soils or sediments may be more restrictive and will result in a larger sphere of influence around the sampling point. While the minimum volume required for measurement using the MPS is

small (~7 mL) and comparable to other small volume sediment samplers (e.g. Rhizon in situ samplers, Seeberg-Elverfeldt et al., 2005), bi-directional pumping from multiple sources by the MPS requires closer to 10-15 mL to reduce cross contamination for accurate solute measurement. Sampling of fine soils with low hydraulic conductivity using the MPS can be aided by the use of small diameter sampling wells similar to those described in Sec. 3.1."

6.Many typos such as "PTFE", "through" should be corrected.

Response : corrected several typos of "though" instead of "through". The authors are unclear for the typo of PFTE, as this is the commonly used abbreviation for polytetrafluoroethylene tubing.

---

## Author Response (AR1)

[revised manuscript text omitted]

**2.1 System dDesign and hHardware**

**Commented [BM3]:** Per request of editor : reconsider the sub-headings in Section 1 and 5

**Commented [BM4]:** Response to comment 3 from RC1 : "Multiplexing" can mean different…"

**Commented [BM5]:** Response to comment 4 from RC1

[revised manuscript text omitted]

**Commented [BM12]:** Per request of the editor : reconside the sub-headings in Section 1 and 5

**Commented [BM13]:** Response to comment 5 & 6 from RC "If one of the challenges is tube clogging……." And comment # from RC2

[revised manuscript text omitted]

**Commented [BM15]:** In response to comment 9 from RC 1 Log scale did not improve visibility, decided to change figure to have less time series on a single plot, divided into 3 separate plot

[revised manuscript text omitted]

10    **Figure 2: The logo of Copernicus Publications.**